# Benefits of Pulmonary Rehabilitation in Patients with Idiopathic Pulmonary Fibrosis Receiving Antifibrotic Drug Treatment

**DOI:** 10.3390/jcm11185336

**Published:** 2022-09-11

**Authors:** Yuji Iwanami, Kento Ebihara, Keiko Nakao, Naofumi Sato, Midori Miyagi, Yasuhiko Nakamura, Susumu Sakamoto, Kazuma Kishi, Sakae Homma, Satoru Ebihara

**Affiliations:** 1Department of Rehabilitation Medicine, Toho University Omori Medical Center, 6-11-1 Omori-nishi, Ota-ku, Tokyo 143-8541, Japan; 2Department of Respiratory Medicine, Toho University School of Medicine, 6-11-1 Omori-nishi, Ota-ku, Tokyo 143-8541, Japan; 3Department of Advanced and Integrated Interstitial Lung Diseases Research, Toho University School of Medicine, 6-11-1 Omori-nishi, Ota-ku, Tokyo 143-8541, Japan; 4Department of Internal Medicine and Rehabilitation Science, Tohoku University Graduate School of Medicine, 1-1 Seiryo-machi, Aoba-ku, Sendai 980-8574, Japan

**Keywords:** antifibrotic drugs, pulmonary rehabilitation, idiopathic pulmonary fibrosis

## Abstract

Background: Although patients with idiopathic pulmonary fibrosis (IPF) often receive treatment with antifibrotic drugs (AFDs) and pulmonary rehabilitation (PR) concurrently, there are no reports on the effect of PR on patients with IPF receiving AFDs. Therefore, we investigated the effect of PR on patients with IPF receiving AFDs. Methods: Eighty-seven eligible patients with IPF (61 male; 72.0 ± 8.1 years; GAP severity stage I/II/III: 26/32/12) were recruited for the study. Patients who completed a 3-month outpatient PR program and those who did not participate were classified into four groups according to use of AFDs: PR group (n = 29), PR+AFD group (n = 11), treatment-free observational group (control group; n = 26), and AFD group (n = 21). There was no significant difference in age, sex, or severity among the groups. Patients were evaluated for physical functions such as 6-min walk distance (6MWD) and muscle strength, dyspnea, and health-related quality of life (HRQOL) at baseline and at 3 months. Results: In the PR group, dyspnea and 6MWD showed significant improvement after the 3-month PR program (*p* < 0.05 and *p* < 0.01, respectively). HRQOL was significantly worse at 3 months (*p* < 0.05) in the AFD group, but not in the other groups. The change in 6MWD from baseline to the 3-month time point was significantly higher in the PR+AFD group than in the AFD groups (*p* < 0.01). Conclusions: It was suggested that AFD treatment reduced exercise tolerance and HRQOL at 3 months; however, the concurrent use of PR may prevent or mitigate these effects.

## 1. Introduction

Idiopathic pulmonary fibrosis (IPF) is an irreversible, chronic, and progressive disease with a poor prognosis due to severe fibrosis [1,2]. As IPF progresses, it leads to decreased forced vital capacity (FVC), severe hypoxemia, impaired exercise tolerance, and reduced health-related quality of life (HRQOL) [3]. The antifibrotic drugs (AFDs) pirfenidone and nintedanib are recommended for the treatment of IPF [2,4,5,6].

AFDs suppress the decline in FVC and progression of the disease [4,5,6,7]. However, AFDs have not been reported to improve IPF-associated dyspnea, impaired exercise tolerance, and reduced HRQOL. Therefore, other nonpharmacological interventions are important for managing the symptoms experienced by patients with IPF [8]. One of the pillars of nonpharmacological therapy is pulmonary rehabilitation (PR). A Cochrane meta-analysis on patients with interstitial lung disease (ILD) showed that PR improves dyspnea, exercise tolerance, and HRQOL [9]. However, the effect of PR on indicators of lung function, such as predicted forced vital capacity (%FVC) and lung diffusion capacity for carbon monoxide was limited [10].

AFDs and PR are recommended in national and international guidelines [1,8]. Therefore, patients with IPF often receive AFD treatment and PR concurrently, but no study has reported the effect of PR on patients with IPF receiving AFDs. AFDs and PR have different mechanisms of action and may have a synergistic effect. PR is the only treatment that improves exercise tolerance and HRQOL, but its effectiveness is limited in advanced stages of the disease [11]. Therefore, concomitant use of AFDs, which suppress disease progression, may enhance the effect of PR and compensate for its shortcomings.

Side effects have been reported to occur in approximately 60% and 30% of patients receiving nintedanib and pirfenidone, respectively [6,12]. Therefore, there is concern that side effects (e.g., photosensitivity reaction, diarrhea, anorexia, fatigue from liver dysfunction) of AFDs may limit daily life and lead to a decline in HRQOL and 6-min walk distance (6MWD) [13]. Consequently, we investigated the effect of PR on patients with IPF receiving AFDs.

## 2. Materials and Methods

### 2.1. Study Design

This study analyzed data from a subgroup of 114 patients with IPF among the participants of the Toho Rehabilitation for Interstitial Pneumonia (TRIP) study enrolled between July 2014 and February 2019 [14]. The TRIP study is an ongoing project at the Toho University Medical Center Omori Hospital (Tokyo, Japan) to assess the long-term effects of PR in patients with interstitial lung disease (ILD) through a two-year follow-up. The study was approved by the Ethics Committee of Toho University Omori Medical Center (approval 27–82 and M21153). The study was registered with the authorized clinical trial registry of International Committee of Medical Journal Editors (UMIN Clinical Trials Registry: UMIN000047241). Written informed consent for participation was obtained from all patients before enrollment in the study. This is a prospective, nonrandomized, controlled observational study in which stable patients with ILD undergo a 3-month outpatient pulmonary rehabilitation program (PRP). Participation in the PRP is elective, and if a patient chooses not to participate, evaluation is performed at baseline and 3 months later.

### 2.2. Participants

Of the 164 patients who were recruited in the TRIP study between July 2014 and February 2019, 114 patients who were diagnosed with IPF, medically stable, and able to walk independently were included in this study. IPF was diagnosed in accordance with the American Thoracic Society/European Respiratory Society statement via multidisciplinary discussion [1].

Exclusion criteria were as follows: patients with orthopedic or central nervous system disorders that cause gait disturbance, dementia, and other diseases with poor prognosis (e.g., terminal malignant tumors, severe heart failure). Further, patients who had participated in PRP at least once in the past or changed medications within the past 3 months, including AFDs, were excluded.

Patients who completed outpatient PRP (once a week for 3 months) and those who did not participate were classified into four groups according to use of AFDs: PR group (*n* = 29), PR+AFD group (*n* = 11), treatment-free observational group (control group, n = 26), and AFD group (n = 21) (Figure 1).

AFD treatment was appropriately commenced by the treating pulmonologist once IPF was diagnosed. Some patients received AFDs before the start of rehabilitation and some started both simultaneously.

### 2.3. Pulmonary Rehabilitation Program

Participants underwent a weekly PR session for three months on an outpatient basis, and each session lasted 60 min. PRP consisted of aerobic exercise, upper and lower limb resistance training, breathing exercises (pursed lip breathing, diaphragmatic breathing), trunk-centered stretching, and patient education (disease knowledge, self-management of disease, and how to use oxygen). PRP was individually supervised by a physical therapist.

Aerobic exercise was performed using a treadmill at 60–80% of the patient’s maximum walking speed for at least 20–30 min. The maximum walking speed was calculated based on the 6MWD at baseline. Weights were used for resistance training of the upper and lower limbs. Each set consisted of 10–15 repetitions, and the volume was gradually increased to 3–5 sets.

The PR group was also instructed to perform aerobic exercise, including resistance training, limb/trunk stretching, and walking, at least 2–3 days a week as a home exercise program. Patients were also asked to record whether they had a home exercise program. In addition, physical activity (number of steps by the pedometer), degree of dyspnea, and vital signs (blood pressure, pulse and SpO_2_ as measured by the patients) were recorded daily to improve self-management ability. The recorded content was confirmed and discussed by the physiotherapist at the outpatient visits.

For patients who did not wish to have outpatient PR (control and AFD groups), we provided guidance, using a pamphlet on how to exercise at home, at the initial evaluation.

### 2.4. Measurements

Patients were evaluated at baseline and at 3 months. At baseline, medication history, use of oxygen therapy, and respiratory function were evaluated. Vital function tests including FVC, forced expiratory volume in 1 s (FEV_1_), FEV_1_/FVC, and lung diffusion capacity for carbon monoxide were measured according to guidelines [15,16]. Arterial blood gas analysis was conducted using a spectrophotometer (ABL800 Flex; Radiometer Medical) on arterial blood collected at rest. The severity of IPF was assessed using a multidimensional index and staging system (Sex-Age-Physiology Index Stage) [17].

Dyspnea, HRQOL, 6MWD, quadriceps force (QF), and hand grip strength (HGS) were assessed at baseline and 3 months later. Subjective dyspnea was assessed using the modified Medical Research Council dyspnea scale (mMRC scale) and graded from 0 to 4 [18]. Quadriceps force (QF) was measured using a hand-held dynamometer (a Mobie: Sakai Medical Corp., Tokyo, Japan) to measure isometric knee extension muscle [19]. Patients sat on a training bench and adjusted the position of their gluteal region so that a bench leg was behind the lower extremity on the measurement side. Measurements were performed three times for each leg at intervals of 30 s and the largest value was used to calculate the ratio of knee extension strength to body weight. Hand grip strength (HGS) was measured using a hand dynamometer in the standing, supinated position. HGF was assessed for each hand with the shoulder and wrist in neutral position. Measurements were performed three times for each hand and the largest value was used as HGF.

HRQOL was evaluated using the chronic obstructive pulmonary disease (COPD) Assessment test (CAT) and St. George’s Respiratory Questionnaire (SGRQ) scores [20,21]. CAT includes a simple questionnaire consisting of eight questions. Each item is evaluated from 0 to 5, and the higher the score, the worse the health condition. The SGRQ consists of 72 questions, and the higher the score, the lower the HRQOL. Both questionnaires were initially developed for patients with COPD but have been shown to be effective in the evaluation of patients with IPF [22,23].

The 6-min walk test was conducted in accordance with the American Thoracic Society guidelines [24]. Patients on long-term oxygen therapy were tested at the oxygen flow rate during exertion, as directed by their physician. The ratio of the predicted values was calculated using the prediction formula of Enright et al. (%6MWD) [25].

### 2.5. Statistical Analysis

Baseline values were subtracted from the 3-month values for 6MWD, %6MWD, QF, HGS, mMRC, and SGRQ and CAT scores, and the change from baseline to the 3-month time point (delta [Δ]) was calculated.

The Wilcoxon rank-sum test was used for comparison between values at baseline and at the 3-month time point because the data were not normally distributed based on the estimation using the Shapiro–Wilk test. One-way analysis of variance (one-way ANOVA), multiple comparisons (post hoc Tukey’s test), and χ^2^ tests were used for between-group comparisons of changes from baseline. Spearman’s correlation coefficient was used to assess the relationship between Δ6MWD, ΔmMRC, and ΔSGRQ. Statistical significance was set at *p* < 0.5. All analyses were conducted using SPSS ver.17 (SPSS Inc., Chicago, IL, USA).

## 3. Results

Sixty participants requested PR and 54 patients were not interested in participating in PR. Of the 60 patients, 40 completed PRP, and we successfully followed-up 47 patients who did not participate in PR.

Table 1 shows the patient characteristics at baseline. There were no significant differences between the four groups with regard to age, body mass index, use of oxygen therapy, severity, and lung function other than FEV_1_/FVC.

Table 2 shows the results of the mMRC, 6MWD, HGS, QF, and HRQOL (SGRQ and CAT scores) at baseline and at 3 months. As for mMRC, the PR group showed significant improvement (*p* < 0.05) whereas other groups did not. Similarly to the 6 min walk test, the PR group showed significant improvement in both 6MWD (*p* < 0.01) and %6MWD (*p* < 0.01), whereas other groups did not. In muscle strength, there were no significant changes in any group. As for HRQOL, the CAT score was significantly worse at 3 months compared to that at baseline (*p* < 0.05) in the AFD group, but not in the other groups.

Table 3 shows the changes in mMRC, 6MWD, HGS, QF and HRQoL over the 3-month intervention period. A comparison of the four groups using one-way ANOVA showed that ΔmMRC was significantly worse in the AFD group than in the PR group (*p* < 0.05) while this was not the case for the other groups. There were no significant differences between groups in other variables (Table 3).

The changes in 6MWD and %6MWD are shown in Figure 2A,B, respectively. Δ6MWD was significantly higher in the PR group than in the control and AFD groups (*p* < 0.05 and *p* < 0.001, respectively). In addition, the values in the PR+AFD group were significantly higher than those in the AFD group (*p* < 0.01). Similarly, Δ%6MWD was significantly higher in the PR group than in the control and AFD groups (*p* < 0.05 and *p* < 0.001 respectively). Moreover, the values in the PR+AFD group were significantly higher values than those in the AFD group (*p* < 0.05).

Further, we investigated the association between Δ6MWD and ΔmMRC and ΔSGRQ in all patients. A significant negative correlation was found between ΔmMRC and Δ6MWD/Δ%6MWD (*r* = −0.337, *p* < 0.05 and *r* = −0.331, *p* < 0.05, respectively, Figure 3A,B). In addition, Δ6MWD and Δ%6MWD showed a significant negative correlation with ΔSGRQ (*r* = −0.277, *p* < 0.05 and *r* = −0.301, *p* < 0.05, respectively; Figure 3C,D).

Table 4 shows the AFD administration period at baseline in the PR+AFD and AFD groups, frequency of side effects, and their breakdown. There were no significant differences in the duration of AFD administration, incidence of overall side effects, or incidence of each side effect between the PR+AFD and AFD groups.

## 4. Discussion

To the best of our knowledge, this is the first study to examine the effects of PR on patients with IPF receiving AFDs. In the AFD group, a decrease in 6MWD was observed, and a significant improvement in Δ6MWD was observed in the PR+AFD group compared to the AFD group, suggesting a beneficial effect of PR on patients receiving AFDs.

In recent systemic reviews, PR was shown to improve exercise tolerance (6MWD and peak volume of oxygen consumed), reduce dyspnea, and improve HRQOL in patients with IPF [9,10]. In our study, the PR group showed a significant increase in 6MWD compared to the control group, which is consistent with the findings in previous studies [9]. It is also consistent with previous findings that the extended distance was about 40 m [9]. In contrast, previous studies on AFDs reported that pirfenidone suppressed the decline in FVC, 6MWD, and dyspnea scores after 52 weeks of intervention [4,12,26]. In addition, nintedanib has been shown to significantly suppress the decrease in FVC compared to placebo after 52 weeks of intervention and, in subgroup analyses, it suppressed deterioration in SGRQ scores in patients with severe IPF [6,13]. However, no study has reported the effects of AFDs after administration for only 3 months [4,13]. Therefore, in this study, we investigated the effect of PR with concurrent AFD administration and AFD monotherapy on 6MWD and HRQOL at 3 months.

An important consideration in AFD administration is side effects. The most common side effects are gastrointestinal symptoms, such as nausea, diarrhea, loss of appetite, and liver damage. These symptoms are likely to impair activities of daily living and HRQOL in patients with IPF [13]. In addition, pirfenidone has been reported to cause photosensitivity and rashes, which may limit outdoor activities of daily life and affect exercise tolerance and quality of life [6].

In this study, side effects similar to those in a previous study were observed in the AFD and PR+AFD groups (approximately 40% and 60% of participants, respectively), but no patient required discontinuation of medication due to side effects. However, in the AFD group, an average decrease of approximately 35 m in 6MWD was observed at 3 months from baseline. Since the minimum clinically significant difference in 6MWD in patients with IPF is 28 m, this change is clinically relevant, despite not being significant [27]. This suggests that AFD use affected 6MWD. In contrast, Δ6MWD in the PR+AFD group was significantly higher than that in the AFD group. This suggests that PR can prevent the decrease in 6MWD caused by AFDs in patients with IPF.

In this study, mMRC scale scores, which are an index of shortness of breath, showed a significant improvement after PR in the PR group. In addition, a significant difference was found in ΔmMRC values between the AFD and PR groups. Thus, the findings suggest that PR improves dyspnea.

Regarding the effect of AFDs on dyspnea, Kreuter et al. used the University of California San Diego shortness of breath questionnaire (UCSD-SOBQ) and reported that AFDs suppressed deterioration in the UCSD-SOBQ score compared to that with a placebo. However, this effect was not sufficient to improve dyspnea [13,28]. Therefore, it was suggested that PR is important for improving dyspnea. In the present study, ΔmMRC was found to have a significant negative correlation with Δ6MWD and Δ% 6MWD (Figure 3A,B). Thus, improvement in dyspnea contributes to improvement in the 6MWD.

In contrast, no significant improvement in dyspnea was observed in the PR+AFD group. This may be due to the small sample size of the PR+AFD group.

HRQOL is an important outcome of the effect of PR on IPF. In this study, SGRQ and CAT scores showed no significant difference in HRQOL between the groups. A previous study showed that compared to a placebo, AFDs significantly suppressed the deterioration of HRQOL assessed using SGRQ scores [13]. However, no significant change observed in the present study may be attributed to the short study period of 3 months (12 weeks) compared to the 52-week intervention in the previous study. Quality of life is influenced by a variety of factors such as exercise performance, daily symptoms and emotional factors, which can lead to variability in data [29]. In this study there was wide variability of change in SGRQ scores, which may have prevented detection of significant differences. In contrast, significant deterioration in CAT scores was observed only in the AFD group. It is possible that AFD can worsen quality of life at 3 months, as this is when side effects are likely to occur.

We also investigated the relationship between ΔSGRQ, Δ6MWD, and Δ% 6MWD in patients with IPF and found a significant negative correlation (Figure 3 C,D). In a previous cross-sectional study on the relationship between 6MWD and HRQOL, Verma et al. showed a significant correlation between SGRQ and 6MWD in 82 patients with IPF [30]; however, our study was longitudinal. Regarding the amount of change observed over three months, the fact that improvement in 6MWD contributed to the improvement of the SGRQ score was a new finding.

In a subanalysis of a larger multicenter study, patients with IPF under AFD treatment showed a trend for higher improvement in exercise capacity as compared to those not treated, suggesting a synergistic effect of AFD and PR. In contrast, our result showed PR tended to counteract the adverse effects of AFD on 6WMD [31]. This might be due to racial differences and differences in the frequency and duration of PR. Further study is warranted to clarify these points.

This study had some limitations. First, the sample was small, thus making the power of our inferences low. Using the mean and standard deviation data provided in our previous TRIP study report [14], the calculated sample sizes with power of 80% to detect significant change were 22 and 15 in each group for 6MWD and %6MWD, respectively. However, since IPF is a rare disease, this obstacle is difficult to overcome, and the sample size used here is not drastically different from that in previous studies. Second, the possibility of bias cannot be ruled out because this study was not randomized, and it was a single-center study.

Third, the large number of dropouts may have affected the results. In this study, the participants were older than those in previous studies. Thus, many patients had difficulty in reaching the hospital. In addition, since IPF is a progressive disease, there were cases of death due to acute worsening of patient condition.

Fourth, nintedanib and pirfenidone were not examined separately. Each has a different mechanism of action and side effects [4,5]; therefore, it is necessary to investigate these AFDs separately in the future. Fifth, we found no difference in the onset of side effects between the AFD group and the PR+AFD group (Table 4). Therefore, it could not be proven that PR suppresses side effects.

Lastly, the length of follow-up was just 3 months in this study. Both AFD and PR are treatments that should be continued as long as possible unless there is a reason to stop. Currently, a randomized controlled study to evaluate the 12-month effects of pulmonary rehabilitation in IPF treated with nintedanib is ongoing [32]. That study may reveal the contribution of concomitant use of nintedanib to the maintenance of long-term effects of pulmonary rehabilitation.

## 5. Conclusions

This study examined the effects of PR with concurrent AFD treatment in patients with IPF. It was suggested that while AFDs reduced exercise tolerance and HRQOL at 3 months, the concurrent use of PR may prevent or mitigate these effects.

## Figures and Tables

**Figure 1 jcm-11-05336-f001:**
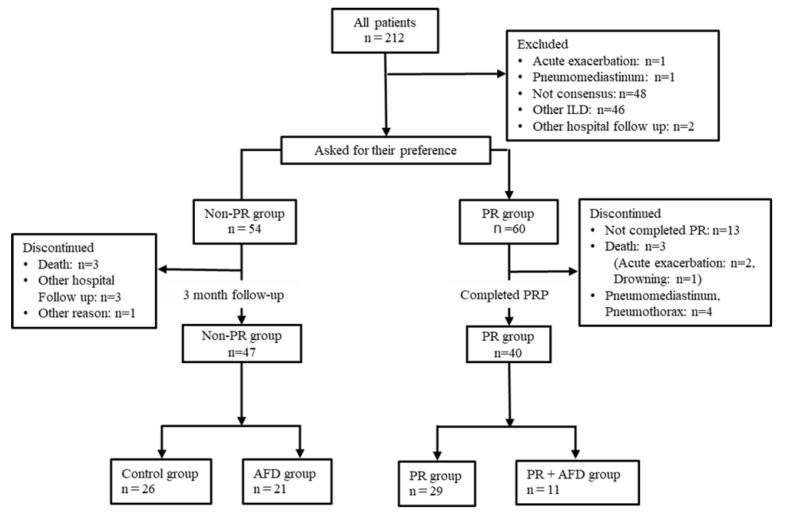
Participant selection flow diagram. Abbreviations: ILD, interstitial lung disease; PR, pulmonary rehabilitation; AFD, antifibrotic drugs.

**Figure 2 jcm-11-05336-f002:**
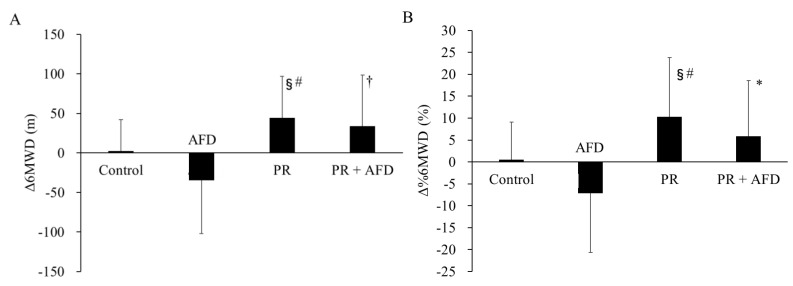
Comparison of the (**A**) change in six-minute walk distance (Δ6MWD) and (**B**) change in the percentage of predicted six-minute walk distance (Δ%6MWD) at 3 months between groups. Abbreviations: PR, pulmonary rehabilitation; AFD, antifibrotic drugs; 6MWD, six-minute walk distance; %6MWD, percent predicted six-minute walk distance. Vertical columns indicate means ± SD. *p* values were calculated using a one-way analysis of variance with post hoc Tukey’s test. §: *p* < 0.05 vs. control, #: *p* < 0.001 vs. AFD, †: *p* < 0.01 vs. AFD, *: *p* < 0.05 vs. AFD.

**Figure 3 jcm-11-05336-f003:**
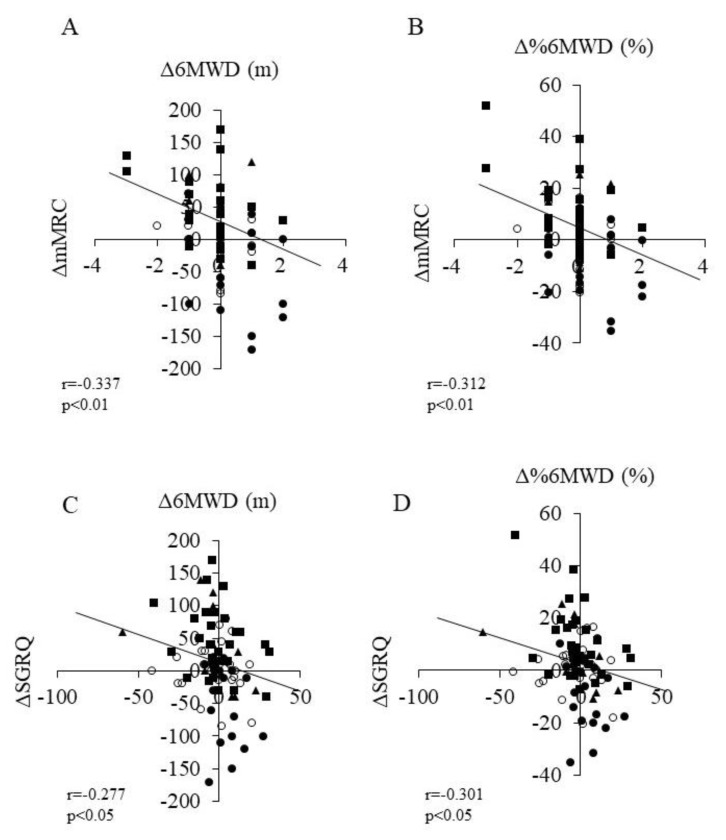
Relationship between (**A**) Δ6MWD and ΔmMRC scale scores, (**B**) Δ%6MWD and ΔmMRC scale scores, (**C**) Δ6MWD and ΔSGRQ scores, and (**D**) Δ%6MWD and ΔSGRQ scores in all patients. ○: control group, ■: PR group, ●: AFD group, ▲: PR+AFD group. Abbreviations: AFD, antifibrotic drugs; PR, pulmonary rehabilitation; Δ, delta; 6MWD, six-minute walk distance; mMRC, modified Medical Research Council; SGRQ, St George Respiratory Questionnaire.

**Table 1 jcm-11-05336-t001:** Baseline characteristics of each group.

	All (*n* = 87)	Control (*n* = 26)	AFD (*n* = 21)	PR (*n* = 29)	PR+AFD (*n* =11)	*p* Value
Age, year	72.0 ± 8.1	73.0 ± 8.5	70.3 ± 6.6	73.5 ± 8.9	69.0 ± 7.4	0.29 ^a^
Sex, male/female	61/26	14/12	18/3	18/11	11/0	<0.05 ^b^
BMI, kg/m^2^	22.9 ± 3.9	24.0 ± 3.6	22.3 ± 4.2	22.5 ± 3.9	22.5 ± 4.1	0.38 ^a^
Oxygen use (%)	21 (30)	3 (15)	6 (40)	9 (33)	3 (37.5)	0.34 ^b^
Smoking status (current/former/never)	1/44/25	1/13/12	0/14/7	0/19/10	0/9/2	0.47 ^b^
Medications						
Azathioprine	0	0	0	0	0	
Cyclophosphamide	0	0	0	0	0	
Cyclosporine	2	1	0	1	0	
Tacrolimus	0	0	0	0	0	
Prednisolone	19	5	6	3	4	
Pirfenidone	15	0	10	0	5	
Nintedanib	17	0	11	0	6	
N-Acetylcysteine	18	4	4	6	2	
Severity						
GAP stage	1.6 ± 0.7	1.4 ± 0.6	1.6 ± 0.7	1.7 ± 0.7	2.0 ± 0.7	0.15 ^a^
(I/II/III)	26/32/12	16/8/2	10/8/3	11/13/5	3/5/3	
Pulmonary function						
FVC, % pred	77.1 ± 20.5	84.2 ± 18.2	71.6 ± 15.7	77.9 ± 23.2	68.5 ± 22.1	0.08 ^a^
FEV_1_, % pred	91.2 ± 22.6	95.7 ± 18.6	85.5 ± 17.3	95.1 ± 28.6	81.5 ± 19.5	0.15 ^a^
FEV_1_/FVC, %	86.1 ± 13.2	79.9 ± 8.0 ^c^	90.5 ± 14.5	87.7 ± 15.7	88.3 ± 9.4	<0.05 ^a^
DL_CO_, % pred	60.1 ± 21.1	69.6 ± 22.4	55.8 ± 22.1	57.8 ± 18.1	53.1 ± 18.9	0.056 ^a^
PaO_2_, torr	81.9 ± 13.4	86.1 ± 13.2	84.1 ± 14.2	77.2 ± 12.0	79.9 ± 13.8	0.07 ^a^
PaCO_2_, torr	41.7 ± 5.2	41.2 ± 4.1	41.4 ± 3.6	41.7 ± 6.7	43.7 ± 5.6	0.61 ^a^
IP marker						
KL-6, U/mL	1040.3 ± 743.5	790.8 ± 428.0	1185.8 ± 921.6	1098.9 ± 759.5	1198.0 ± 859.2	0.22 ^a^
SP-A, U/mL	68.3 ± 30.9	67.6 ± 33.7	68.5 ± 36.5	67.5 ± 27.8	71.7 ± 22.7	0.98 ^a^
SP-D, U/mL	239.5 ± 163.6	176.7 ± 102.5	258.5 ±108.6	282.4 ± 236.9	239.1 ± 82.6	0.10 ^a^

Data are reported as means ± SD or number (n). ^a^ *p* values calculated using a one-way analysis of variance with post hoc Tukey test. ^b^ *p* values calculated using a χ^2^ test. ^c^ *p* < 0.05 vs. AFD. Abbreviations: n.s., not significant; AFD, antifibrotic drug; PR, pulmonary rehabilitation; BMI, body mass index; JRC, the Japanese Respiratory Society IPF disease severity classifications; GAP stage, Sex-Age-Physiology Index Stage; FVC, forced vital capacity; % pred, percent predicted; FEV_1_, forced expiratory volume in 1 s; DLCO, lung diffusion capacity for carbon monoxide; IP marker; interstitial pneumonia serum marker, KL-6, Krebs von den Lungen-6; SP-A, surfactant protein-A; SP-D, surfactant protein-D.

**Table 2 jcm-11-05336-t002:** Comparison of results at baseline and 3 months in each group.

	Control		AFD		PR		PR+AFD	
	Baseline	3 Months	Baseline	3 Months	Baseline	3 Months	Baseline	3 Months
mMRC scale	1.3 ± 1.1	1.2 ± 0.9	1.4 ± 1.0	1.7 ± 1.2	1.5 ± 1.0	1.1 ± 0.8 ^a^	1.9 ± 1.1	1.7 ±1.1
6 min walk test								
6MWD (m)	410.5 ± 98.1	413.2 ± 112.3	409.2 ± 115.1	374.7 ± 155.2	365.8 ± 96.9	410.5 ± 107.1 ^b^	386.3 ± 91.1	420.4 ±124.4
% 6MWD (%)	86.0 ± 21.6	86.6 ± 24.8	83.1 ± 21.2	76.0 ± 30.4	73.5 ± 20.2	84.1 ± 23.7 ^b^	78.5 ± 16.5	84.5 ±19.1
Muscle strength								
QF (Nm/kg)	1.2 ± 0.4	1.2 ± 0.5	1.3 ± 0.3	1.4 ± 0.5	1.3 ± 0.4	1.3 ± 0.3	1.3 ± 0.5	1.3 ±0.6
HGS (kg)	24.0 ± 7.8	24.2 ± 7.8	29.1 ± 7.3	28.5 ± 7.0	26.2 ± 8.3	26.4 ± 8.0	29.4 ± 7.7	30.2 ±7.8
HRQOL								
SGRQ total	32.3 ± 20.1	30.3 ± 16.5	38.3 ± 21.1	42.9 ± 20.4	39.6 ± 18.5	40.0 ± 20.2	52.1 ± 20.2	52.0 ±20.2
CAT	11.8 ± 9.1	11.8 ± 7.5	13.2 ± 9.1	15.9 ± 7.2 ^c^	14.0 ±7.4	14.5 ± 7.4	20.1 ± 9.5	21.5 ±10.2

Data are reported as means ± SD. *p* values were calculated using the Wilcoxon rank-sum test. ^a^ *p* < 0.05 vs. PR at baseline; ^b^ *p* < 0.01 vs. PR at baseline; ^c^ *p* < 0.05 vs. AFD at baseline. Abbreviations: AFD, antifibrotic drugs; PR, pulmonary rehabilitation; mMRC, modified Medical Research Council test; 6MWD, six-minute walk distance; % 6MWD, percent predicted six-minute walk distance; QF, quadriceps force; HGS, hand grip strength; HRQOL, Health-Related Quality of Life; SGRQ, St George Respiratory Questionnaire; CAT, Chronic obstructive pulmonary disease assessment test.

**Table 3 jcm-11-05336-t003:** Comparison of the change between baseline and 3 months between the groups.

	Control	AFD	PR	PR+AFD
Dyspnea				
ΔmMRC scale	−0.07 ± 0.6	0.3 ± 0.9 ^a^	−0.4 ± 0.9	−0.1 ± 0.6
	(−0.3 to 0.2)	(−0.1 to 0.7)	(−0.7 to −0.06)	(−0.5 to 0.2)
Peripheral muscle strength				
ΔQF (Nm/kg)	0.06 ± 0.2	0.1 ± 0.4	−0.01 ±0.3	−0.04 ± 0.4
	(−0.03 to 0.1)	(−0.08 to 0.3)	(−0.1 to 0.1)	(−0.3 to 0.3)
ΔHGS (kg)	0.2 ± 2.8	−0.5 ± 2.8	0.2 ± 2.1	0.6 ± 1.5
	(−0.9 to 1.3)	(−1.8 to 0.7)	(−0.5 to 1.0)	(−0.2 to 1.8)
HRQOL				
ΔSGRQ total	−1.9 ± 13.7	4.5 ± 9.6	0.4 ± 14.9	−0.04 ± 12.5
	(−7.5 to 3.6)	(0.7 to 8.9)	(−5.2 to 6.0)	(−8.4 to 8.4)
ΔCAT	−0.03 ± 6.2	2.6 ± 5.6	0.5 ± 6.6	1.3 ± 7.6
	(−2.5 to 2.4)	(0.04 to 5.1)	(−2.0 to 3.0)	(−3.7 to 6.5)

Data are reported as means ± SD. (95% confidence interval). *p* values were calculated using a one-way analysis of variance post hoc Tukey’s test. ^a^ *p* < 0.05 vs. PR. Abbreviations: AFD, antifibrotic drugs; PR, pulmonary rehabilitation; Δ, delta; mMRC, modified Medical Research Council; QF, quadriceps force; HGS, hand grip strength; HRQOL, Health-Related Quality of Life; SGRQ, St George Respiratory Questionnaire; CAT, COPD assessment test.

**Table 4 jcm-11-05336-t004:** Treatment adverse events in the AFD and PR+AFD groups.

	AFD	PR+AFD	*p* Value
Duration of AFD treatment at baseline, (days)	6.9 ±13.4	6.2 ±15.2	0.85 ^a^
Number of patients with adverse events, *n* (%)			0.37 ^b^
(−)	12 (57.1)	4 (36.3)	
(+)	9 (42.8)	7 (63.6)	
Adverse events			
Nausea, *n* (%)	0	1 (9.0)	0.16 ^b^
Diarrhea, *n* (%)	5 (23.8)	3 (27.2)	0.37 ^b^
Liver dysfunction, *n* (%)	3 (14.2)	1 (9.0)	0.67 ^b^
Decreased appetite, *n* (%)	2 (9.5)	3 (27.2)	0.18 ^b^
Fatigue, *n* (%)	0	1 (9.0)	0.16 ^b^
Dizziness, *n* (%)	0	1 (9.0)	0.16 ^b^
Photosensitivity reaction, *n* (%)	1 (4.7)	0	0.46 ^b^
Thrombocytopenia, *n* (%)	0	1 (9.0)	0.16 ^b^

Data are reported as means ± SD or number (%). ^a^ *p* values calculated using Mann–Whitney U test. ^b^ *p* values calculated using χ^2^ test. Abbreviations: n.s., not significant; AFD, antifibrotic drugs; PR, pulmonary rehabilitation.

## Data Availability

The data presented in this study are available on request from the corresponding author.

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
