# Peer review of "Benefits of Pulmonary Rehabilitation in Patients with Idiopathic Pulmonary Fibrosis Receiving Antifibrotic Drug Treatment"

_jcm, 2022, doi:10.3390/jcm11185336_

Round 1

Reviewer 1 Report

General comments to author:

This is a novel and well written manuscript looking at the effect of PR in patients receiving antifibrotic drug treatment, concluding that the concurrent use of pulmonary rehabilitation with antifibrotic drug treatment may prevent or mitigate the declines shown in exercise tolerance and HRQoL following 3 months of antifibrotic drug treatment alone. The authors provide a clear rationale for the study and whilst the study is not randomised, the authors acknowledge this as a limitation and describe most of the methodology and data collection procedures well. Overall, I have minor comments for the author to address as detailed below.

Specific Comments: 

Abstract:

- Line 20: It would be useful to add details of how many patients you included in the study and basic characteristics such as age, gender, disease severity etc.

- Line 23: Physical functions is quite vague, can the authors add more detail of specific variables here?

Introduction:

-       The introduction is well written and provides a clear rationale for the study.

Methods:

2.1 Study Design

- Line 65: There is a small typo, I think it should read as “This study analysed..”

- Was the TRIP study protocol pre-registered on clinicaltrials.gov? I would suggest adding the clinical trials reference number for the study protocol and details of ethical approval.

2.2 Participants

- I would recommend adding a sentence about the informed consent process.

 2.4 Measurements:

- The outcome measures are outlined clearly, however could you provide more detail on how quadriceps force and hand grip strength were measured. For example, was it the best of 3 measurements that were taken? Was it the patient’s dominant hand/leg that was used?

- Line 50: There is a small typo here. I think it should read as “but has been shown” instead of “sown”.

2.5 Statistical Analysis

- The Wilcoxon rank sum test was used to compare values at baseline to 3 months. Could you explain why this non-parametric test was used? Was it due to data being not normally distributed? It would be good if the authors could add some detail about whether the normality of data was checked.

Results:

- Figure 1: In the flow diagram, authors have split the groups for PR and non-PR. I would suggest including the number of patients for all four groups (Control, AFD, AFD+PR and PR) at the start and end of the intervention, to appreciate the rates of consent and completion in each of the four groups.  

- Lines 194 and 195: The authors state three outcomes (mMRC, 6MWD and %6MWD), however only two P values are stated. Could you please include the P values for all three outcomes.

- Line 193 to 197: The authors have outlined the variables and groups that showed significant improvements over the 3 months. For CAT scores, the authors outline the significant result in the AFD group, whilst also stating what happened in the other groups for this variable. I would also recommend doing this for mMRC, 6MWD and %6MWD, as only the PR group is described.

- Table 2: I would recommend changing the table title to “Comparison of results at baseline and 3 months in each group” or “pre and post intervention”, as not all groups received PR.

- Line 212 to 213: I think it would be clearer to say, “Table 3 shows the changes in mMRC, 6MWD, HGS, QF and HRQoL over the 3-month intervention period.”

- Line 214 to 215: The authors describe that change in mMRC scores is significantly worse in the AFD group compared to the PR group. To make it clearer, I would suggest adding that this was not the case for the other groups. I would also recommend adding another sentence to say that there were no significant differences between groups in the other variables (QF, HGS…) and refer the reader to the table.

- Line 256 to 258: I’m not sure this is supposed to be included?

Discussion:

- Line 273 to 274: The authors describe how a significant improvement in the PR group was similar to that shown in a previous Cochrane review. I think it would be good to compare the degree of improvement here too (e.g. 40m).

- Line 310 to 312: Are there any references to support the point that the responsiveness of the mMRC could be limited for interventions such as pulmonary rehabilitation?  

- Line 318 to 319: The authors state that quality of life is influenced by a variety of factors. I would suggest providing some examples of these factors.

- Line 331 to 335: Small sample size is identified as a limitation of the study, resulting in low power. How do the authors know that the power of the study is low? Could the authors elaborate on whether any priori sample size calculation was undertaken to determine statistical power? Or if a sample size calculation was not performed then what was the reason for this? 

- From this study it seems that AFDs alone do not improve 6MWD and HRQoL, and the decline shown in these variables may be partly due to the side effects of these treatments. It would be interesting to discuss whether these side effects had any impact on the participation in pulmonary rehabilitation? Were there patients that were unable to complete the intervention due to these side effects?

- The authors could add some discussion about future directions. For instance, a recent protocol has been published that will investigate the long-term effect of pulmonary rehabilitation in patients receiving Nintedanib. This study will address the hypothesis that concomitant use of nintedanib contributes to the maintenance of long-term effects of pulmonary rehabilitation, thus leading to a comprehensive therapeutic approach of “nintedanib and pulmonary rehabilitation” in the antifibrotic era. This could be an interesting discussion point for the current study. 

Reference for protocol: 

Nishiyama, O., Kataoka, K., Ando, M., Arizono, S., Morino, A., Nishimura, K., ... & Kondoh, Y. (2021). Protocol for long-term effect of pulmonary rehabilitation under nintedanib in idiopathic pulmonary fibrosis. ERJ Open Research7(3).

Author Response

We appreciate the time the Reviewers have invested in reviewing our manuscript and for the very favorable comments. For ease of reading, we have organized our responses below by providing the Comment (C) followed by our Response (R). The changes in the revised version of the manuscript are shown using Track Changes functions in Microsoft Word. Please note that the changes made do not influence the content, conclusions, or framework of the paper.

C1: This is a novel and well written manuscript looking at the effect of PR in patients receiving antifibrotic drug treatment, concluding that the concurrent use of pulmonary rehabilitation with antifibrotic drug treatment may prevent or mitigate the declines shown in exercise tolerance and HRQoL following 3 months of antifibrotic drug treatment alone. The authors provide a clear rationale for the study and whilst the study is not randomised, the authors acknowledge this as a limitation and describe most of the methodology and data collection procedures well. Overall, I have minor comments for the author to address as detailed below.

R1: Thank you for your very favorable comments. We have responded accordingly.

Specific Comments:

Abstract:

C2: Line 20: It would be useful to add details of how many patients you included in the study and basic characteristics such as age, gender, disease severity etc.

R2: Accordingly, we have provided details regarding the patients in the Abstract (Line 20–21, 23–25 in the revised manuscript).

C3: Line 23: Physical functions is quite vague, can the authors add more detail of specific variables here?

R3: Accordingly, we have added these details (Line 26–27 in the revised manuscript).

 Introduction:

- The introduction is well written and provides a clear rationale for the study.

Methods:

2.1 Study Design

C3: Line 65: There is a small typo, I think it should read as “This study analysed..”

R3: We have corrected the inadvertent typographical errors (Line 69 in the revised manuscript).

C4: Was the TRIP study protocol pre-registered on clinicaltrials.gov? I would suggest adding the clinical trials reference number for the study protocol and details of ethical approval.

R4: The study was approved by the Ethics Committee of Toho University Omori Medical Center (approval numbers 27-82 and M21153). The study was registered to authorized clinical trial registry of the International Committee of Medical Journal Editors (UMIN Clinical Trials Registry: UMIN000047241). We have added this information (Line 74–77 in the revised manuscript).

2.2 Participants

C5: I would recommend adding a sentence about the informed consent process.

R5: We have added the following sentence: “Written informed consent for participation was obtained from all patients before enrollment in the study” (Line 77–78).

 2.4 Measurements:

C6: The outcome measures are outlined clearly, however could you provide more detail on how quadriceps force and hand grip strength were measured. For example, was it the best of 3 measurements that were taken? Was it the patient’s dominant hand/leg that was used?

R6: We have added the details of measuring quadriceps force and hand grip strength (Line 143–150 in the revised manuscript).

C7: Line 150: There is a small typo here. I think it should read as “but has been shown” instead of “sown”.

R7: We have corrected this inadvertent error (Line 184 in the revised manuscript).

2.5 Statistical Analysis

C8: The Wilcoxon rank sum test was used to compare values at baseline to 3 months. Could you explain why this non-parametric test was used? Was it due to data being not normally distributed? It would be good if the authors could add some detail about whether the normality of data was checked.

R8: The Wilcoxon rank sum test was used because the data were not normally distributed based on the estimation using Shapiro-Wilk test. We have added an explanation (Line 196–197 in the revised manuscript).

Results:

C9: Figure 1: In the flow diagram, authors have split the groups for PR and non-PR. I would suggest including the number of patients for all four groups (Control, AFD, AFD+PR and PR) at the start and end of the intervention, to appreciate the rates of consent and completion in each of the four groups.

R9: Accordingly, we have corrected the figure 1 (Page 3, Figure 1 in the revised manuscript). As for rates of consent, 48 patients without consensus among the initial 212 patients with interstitial pneumonia were patients who did not want to be referred to the Rehabilitation Department, even for the baseline assessment, and were not asked for their preference for a pulmonary rehabilitation program. Therefore, in this study design, we think that it is not really significant to know the rate of consent in each group.

C10: Lines 194 and 195: The authors state three outcomes (mMRC, 6MWD and %6MWD), however only two P values are stated. Could you please include the P values for all three outcomes.

R11: We have corrected this inadvertent error (Line 230–236 in the revised manuscript).

C11: Line 193 to 197: The authors have outlined the variables and groups that showed significant improvements over the 3 months. For CAT scores, the authors outline the significant result in the AFD group, whilst also stating what happened in the other groups for this variable. I would also recommend doing this for mMRC, 6MWD and %6MWD, as only the PR group is described.

R11: Accordingly, we have rewritten this paragraph (Line 230–239 in the revised manuscript).

C12: Table 2: I would recommend changing the table title to “Comparison of results at baseline and 3 months in each group” or “pre and post intervention”, as not all groups received PR.

R12: We have revised the text to “Comparison of results at baseline and 3 months in each group” (Page 6, Line 245, Table 2 in the revised manuscript).

C13: Line 212 to 213: I think it would be clearer to say, “Table 3 shows the changes in mMRC, 6MWD, HGS, QF and HRQoL over the 3-month intervention period.”

R13: We have revised the text accordingly (Line 254–257 in the revised manuscript).

C14: Line 214 to 215: The authors describe that change in mMRC scores is significantly worse in the AFD group compared to the PR group. To make it clearer, I would suggest adding that this was not the case for the other groups. I would also recommend adding another sentence to say that there were no significant differences between groups in the other variables (QF, HGS…) and refer the reader to the table.

R14: Accordingly, we have rewritten the text (Line 259-260 in the revised manuscript).

C15: Line 256 to 258: I’m not sure this is supposed to be included?

R15: We have deleted this text (Line 348–351 in the revised manuscript).

Discussion:

C16: Line 273 to 274: The authors describe how a significant improvement in the PR group was similar to that shown in a previous Cochrane review. I think it would be good to compare the degree of improvement here too (e.g. 40m).

R16: We have added a sentence (Line 371-372 in the revised manuscript).

C17: Line 310 to 312: Are there any references to support the point that the responsiveness of the mMRC could be limited for interventions such as pulmonary rehabilitation?

R17: This was our misunderstanding. In patients with interstitial lung diseases, the mMRC is a reliable tool for assessing breathlessness and symptom severity (Papiris SA, et al. Respir Med. 2005 Jun;99(6):755-61). Therefore, we have deleted the sentence (Line 408–410 in the revised manuscript).

C18: Line 318 to 319: The authors state that quality of life is influenced by a variety of factors. I would suggest providing some examples of these factors.

R18: Based on reference [29], quality of life is influenced by a variety of factors such as exercise performance, daily symptoms, and emotional factors. We have added a phrase (Line 417-418 in the revised manuscript).

C19: Line 331 to 335: Small sample size is identified as a limitation of the study, resulting in low power. How do the authors know that the power of the study is low? Could the authors elaborate on whether any priori sample size calculation was undertaken to determine statistical power? Or if a sample size calculation was not performed then what was the reason for this?

R19: Using the mean and standard deviation data provided in our previous TRIP study report [14], the calculated sample sizes with power of 80% to detect significant change were 22 and 15 in each group for 6MWD and %6MWD, respectively. We have added these details (Line 438-441 in the revised manuscript).

C20: From this study it seems that AFDs alone do not improve 6MWD and HRQoL, and the decline shown in these variables may be partly due to the side effects of these treatments. It would be interesting to discuss whether these side effects had any impact on the participation in pulmonary rehabilitation? Were there patients that were unable to complete the intervention due to these side effects?

R20: Thank you for raising the interesting point. Our physiotherapists are trained to recognize early signs of side effects and report any signs to the patient's pulmonary physician. Probably because their pulmonary physicians consistently responded appropriately to reports, no patients discontinued the pulmonary rehabilitation program in our PR+AFD group. Since this is our speculation, we did not include this content in the manuscript.

C21: The authors could add some discussion about future directions. For instance, a recent protocol has been published that will investigate the long-term effect of pulmonary rehabilitation in patients receiving Nintedanib. This study will address the hypothesis that concomitant use of nintedanib contributes to the maintenance of long-term effects of pulmonary rehabilitation, thus leading to a comprehensive therapeutic approach of “nintedanib and pulmonary rehabilitation” in the antifibrotic era. This could be an interesting discussion point for the current study.

Reference for protocol:

Nishiyama, O., Kataoka, K., Ando, M., Arizono, S., Morino, A., Nishimura, K., ... & Kondoh, Y. (2021). Protocol for long-term effect of pulmonary rehabilitation under nintedanib in idiopathic pulmonary fibrosis. ERJ Open Research, 7(3).

R21: Thank you for pointing out this very interesting issue. Accordingly, we have discussed this issue as a fifth limitation. We also cited the protocol paper (Line 454–459 and reference [32] in the revised manuscript).

Reviewer 2 Report

Benefits of PR in IPF patients have been reported, but certainity of evidence is low to moderate. Therefore, its clear that new studies are needed. In their studies Iwanami et al reported  the effect of PR on patients with IPF receiving antifibrotic drugs. Here is my comments about their article:

General :

1.       Some spelling errors (such as “…. the no significant change …..” line 316.) must be corrected.

 Introduction:

1.       In the second paragraph, (line 39-40) first sentence covers the second one.   

Materials and methods:

1.       Evaluation period is short and sample size is small.

2.       Ä°ts not clear how did the antifibrotic treatment get started? (line 102-105). Once the IPF is diagnosed, the treatment should be started. The authors criterias do not have a scientific basis and should be corrected.

3.       CAT score is generally not used for the evaluation of HRQOL in IPF population.

 Discussion:

11. Authors should discuss the findings of a large, multicenter study which was published in 2021  (Brunetti G, Respiratory Medicine 2021).

Author Response

We appreciate the time the Reviewers have invested in reviewing our manuscript and for the very favorable comments. For ease of reading, we have organized our responses below by providing the Comment (C) followed by our Response (R). The changes in the revised version of the manuscript are shown using Track Changes functions in Microsoft Word. Please note that the changes made do not influence the content, conclusions, or framework of the paper.

General :

C1: Some spelling errors (such as “…. the no significant change …..” line 316.) must be corrected.

R1: In revised version, we carefully checked the manuscript. Further, English language editing was performed by Editage (www.editage.jp) (Line 414 and others in the revised manuscript). 

Introduction:

C2: In the second paragraph, (line 39-40) first sentence covers the second one.

R2: We have deleted the second sentence (Line 43-44 in the revised manuscript). 

Materials and methods:

C3: Evaluation period is short and sample size is small.

R4: We agree that the evaluation period is short and the sample size is small. Therefore, we have discussed these issues as limitations (Line 437–442 and 454–459 in the revised manuscript).

C4: Ä°ts not clear how did the antifibrotic treatment get started? (line 102-105). Once the IPF is diagnosed, the treatment should be started. The authors criterias do not have a scientific basis and should be corrected.

R4: Thank you for pointing out this important issue. We agree that the treatment should be started once IPF is diagnosed. When this research was started in 2014, the evidence for antifibrotic drugs was a bit weak and not well known, so there were many doctors who wanted patients to follow up without AFD. Therefore, we had such criteria at that time. But after a few years, that was no longer the case. Doctors started AFD as soon as IPF was diagnosed. As such, this criterion was obsolete and has been removed (Line 127–131 in the revised manuscript).

C5: CAT score is generally not used for the evaluation of HRQOL in IPF population.

R5: Although they were initially developed for patients with chronic obstructive pulmonary disease, the CAT have been validated in patients with ILD (Nagata K, et al. Respirology. 2012 Apr;17(3):506-12). We have described this (Line 183-185 in the revised manuscript).

Discussion:

C6: Authors should discuss the findings of a large, multicenter study which was published in 2021 (Brunetti G, Respiratory Medicine 2021).

R6: Thank you for bringing this important paper to our attention. Accordingly, we discussed the findings of Brunetti G, Respiratory Medicine 2021 (Line 430-436 in the revised manuscript).